

# Instance attack: an explanation-based vulnerability analysis framework against DNNs for malware detection

Ruijin Sun[1], Shize Guo[2], Changyou Xing[1], Yexin Duan[3], Luming Yang[4], Xi Guo[5] and Zhisong Pan[1]

[1] Army Engineering University of PLA, Nanjing, China
[2] National Computer Network and Information Security Management Center, Beijing, China
[3] Army Military Transportation University of PLA, Zhenjiang, China
[4] National University of Defense Technology, Changsha, China
[5] University of Science and Technology Beijing, Beijing, China

Corresponding authors
Xi Guo, xiguo@ustb.edu.cn
Zhisong Pan, hotpzs@hotmail.com

## ABSTRACT

Deep neural networks (DNNs) are increasingly being used in malware detection and their robustness has been widely discussed. Conventionally, the development of an adversarial example generation scheme for DNNs involves either detailed knowledge concerning the model (*i.e.*, gradient-based methods) or a substantial quantity of data for training a surrogate model. However, under many real-world circumstances, neither of these resources is necessarily available. Our work introduces the concept of the instance-based attack, which is both interpretable and suitable for deployment in a black-box environment. In our approach, a specific binary instance and a malware classifier are utilized as input. By incorporating data augmentation strategies, sufficient data are generated to train a relatively simple and interpretable model. Our methodology involves providing explanations for the detection model, which entails displaying the weights assigned to different components of the specific binary. Through the analysis of these explanations, we discover that the data subsections have a significant impact on the identification of malware. In this study, a novel function preserving transformation algorithm designed specifically for data subsections is introduced. Our approach involves leveraging binary diversification techniques to neutralize the effects of the most heavily-weighted section, thus generating effective adversarial examples. Our algorithm can fool the DNNs in certain cases with a success rate of almost 100%. Instance attack exhibits superior performance compared to the state-of-the-art approach. Notably, our technique can be implemented in a black-box environment and the results can be verified utilizing domain knowledge. The model can help to improve the robustness of malware detectors.

# INTRODUCTION

Malware attack is an important issue in today's cybersecurity community. Thousands of malware attacks are reported every day, according to *Demetrio et al.'s (2021a)* description. Both academia and industry have devoted a lot of manpower to malware detection.

Traditional detection methods, such as support vector machine (SVM) (*Li, Ge & Dai, 2015*) and signature (*Vinod et al., 2012*) require manual feature engineering, which can be a daunting task. Given the vast number of malware instances in existence, the labor-intensive nature of this work renders it both time-consuming and tedious. As deep neural networks (DNNs) have made significant advances in many domains, such as image (*Sharif et al., 2016*) and voice classification (*Qin et al., 2019*), an increasing number of researchers and anti-virus enterprises have begun leveraging DNN-based detectors in the field of cybersecurity. The DNNs models automatically make the classification for malware without expert knowledge. Researchers use deep learning models in an end-to-end manner that operates directly on the raw bytes of Windows Portable Executable (PE) files.

In the domain of cybersecurity, malware detection systems can be broadly classified into dynamic and static approaches. While dynamic systems rely on learning the behavioral features of malware for classification, static systems directly classify files using features without executing them (*Sharif et al., 2019*). This article primarily focuses on the static approach. There exist several byte-based DNN models that have demonstrated comparable performance with traditional methods (*Saxe & Berlin, 2015*; *Raff et al., 2018*). The robustness of the DNNs detection system and the interpretability of DNNs have attracted much attention, while the DNNs have shown great potential. The interpretability of models is particularly important in financial and security-related domains. The absence of model interpretability can significantly limit the applicability of DNN models in these domains. Adversarial examples are the techniques that focus on perturbing the examples to mislead DNN-based detection systems, and can be leveraged to enhance the robustness of such systems. Unlike other domains, semantic invariance constraints must be satisfied in binary. When an adversarial example is generated, its characters may be transformed and its semantic should not be changed. People introduce different transformation techniques that could keep the functionality of the binaries intact (*Anderson et al., 2018*; *Song et al., 2020*; *Park, Khan & Yener, 2019*). In the context of binary-based adversarial attacks, transformations refer to modifications made to a PE file that do not alter the execution of its underlying code (*Anderson et al., 2018*). Despite the considerable progress that has been made in generating adversarial examples for malware detection, there are still a number of unresolved issues. First, only a few articles that use DNNs to detect malware have explained their detection models. The lack of transparency makes it questioned by many people (*Arp et al., 2022*). The uninterpretable model may detect the binaries according to false causalities that are unrelated to any malicious activity (*Arp et al., 2022*). Second, the binary transformation methods used by others focus on the structural part (*Demetrio et al., 2021a*) and the code part (*Sharif et al., 2019*), but they leave the data section alone. Third, after figuring out how to transform malware, they resort to complicated optimization methods (such as genetic algorithms) (*Demetrio et al., 2021a*) or uninterpretable stochastic methods (*Sharif et al., 2019*). These deficiencies limit their performance under the black-box model. Our approach fills the gap.

In this article, we propose the notion of instance-based attacks. Our method is very similar to the transfer-based approach. The most important difference between instance-based and transfer-based methods is that all the data used to train our model is generated

by data augmentation from one single binary. We use an explanation based adversarial example generation technique to test malware detectors by iteratively approximating the decision boundary. Our method is more effective than others in the context of black-box settings. In order to evade the DNNs in fewer steps, we could transform the most influential modules in each round. Furthermore, our optimization method is interpretable and can be verified with domain knowledge. We highlight our contributions below.

- We introduce the concept of the instance-based attack. Rather than training a surrogate model against the entire model, we instead train a surrogate model for an instance, with a specific emphasis on perturbing around that instance. The adversarial instances are generated by iteratively approximating the decision boundary.
- Several prominent detection models are analyzed using a local interpretable model, their characteristics and drawbacks are highlighted. Notably, we observed a lack of focus on data section transformations within PE files, representing a significant gap in current approaches.
- A novel functionality-preserving transformation method is proposed which is suitable for data sections in PE files that have not been evaluated by other authors.
- The theoretical and mathematical foundations of our model are discussed.
- Our method are tested in various scenarios, and the results demonstrate its superiority over other state-of-the-art approaches in black-box settings (*Sharif et al., 2019*). It can achieve a success rate of almost 100% in certain cases.

## BACKGROUND AND RELATED WORK

In this section, we provide an overview of the most commonly used DNN-based malware detection models, with a particular emphasis on their static components. Following this, adversarial methods designed to target the raw bytes of PE files in malware detectors are discussed. Finally, we conclude the section by examining literature that explains the use of DNNs for malware classification.

### DNNs for malware detection

Malware detection plays an important role in the field of cyber security. DNNs have been used widely by researchers in malware classification. The most appealing aspect of the DNNs-based malware detectors is their ability to achieve state-of-the-art performance from raw bits rather than manually crafted features that require tedious human effort. Many DNNs-based detectors have been proposed so far, and we introduce the most famous ones here. *Nataraj et al. (2011)* visualize the malware binaries as gray-scale images. A classification method using standard image features is proposed, based on the observation that malware images belonging to the same family appear very similar in layout and texture. Then they use the classifier originally designed for images to sort malware. *Coull & Gardner (2019)* introduce a DNN with five convolutional and pooling layers. It also has a learnable 10-dimensional embedding layer. At the end of the network, there is a single fully-connected layer and a sigmoid function. *Saxe & Berlin (2015)* employ

four distinct complementary features from the static, benign and malicious binaries. They use a DNNs-based classifier which consists of an input layer, two hidden layers, and an output layer. They translate the output of the DNNs into a score that can be realistically interpreted as an approximation of the probability that the file is malware. *Johns (2017)* proposes deep convolutional neural networks (CNN) that combine a ten-dimensional, learnable embedding layer with a series of five interleaved convolutional and max-pooling layers arranged hierarchically. MalConv (*Raff et al., 2018*) is the most popular CNN model which combines an eight-dimensional trainable embedding layer. *Raff et al. (2018)* have tried many different structures. They have tried deeper networks (up to 13 layers), narrower convolutional filters (width 3–10), and smaller strides (1–10). Finally, they adopted the network consisting of two 1-D gated convolutional windows with 500 strides. We used the MalConv detection model to evaluate the effectiveness of our method.

## Adversarial examples against DNN-based malware detectors

Adversarial examples are the technologies that focus on the minimal input perturbations of break machine learning algorithms. They can expose the vulnerability of the machine learning model and improve the robustness of the DNNs. For example, when DNNs are used in street sign detection, researchers show ways to mislead street signs recognition (*Chen et al., 2019*). Adversarial examples could also fool voice-controlled interfaces (*Qin et al., 2019*), mislead NLP tasks (*Jia & Liang, 2017*). It is natural to introduce adversarial sample techniques to bypass DNNs based malware detectors. However, the semantics of binaries limit the applicability of the existing adversarial methods designed against image, voice, or NLP classifiers transplanted to the cybersecurity realm, because there is a structural interdependence between adjacent bytes. *Anderson et al. (2018)* introduce one way to bypass machine-learning-based detection by manipulating the PE file format. They find several structures in Windows PE files that could be modified without affecting their functionality. *Kreuk et al. (2018)* craft bytes adversarially in regions that do not affect execution. Specifically, they append adversarial bits at the end of files. *Suciu, Coull & Johns (2019)* extend this idea by finding more places to append in PE files, such as in the middle of two sections of PE files. Different padding strategies are also evaluated, including random appending, FGM appending, and benign appending. *Sharif et al. (2019)* manipulate instructions to produce adversarial examples. Instructions are a functional part of binary files. They introduce two families of transformations. The first one named in-Place randomization (IPR) is quoted from *Pappas, Polychronakis & Keromytis (2012)*. The second one named code displacement (Disp) is also adopted in our article as the baseline. Disp relocates sequences of instructions that contain gadgets from their original locations to newly allocated code segments with a *jmp* instruction. *Sharif et al. (2019)* extend the Disp algorithm. They make it possible to displace any length of consecutive instructions, not just those belonging to gadgets. As far as the variable space is concerned, they focus on the structure characteristic or the code characteristic. None of the above-mentioned articles discuss the data segment, although it plays an important role in malware classification.

**Explanation of adversarial machine learning in malware**

In the field of malware, the traditional routine for producing adversarial examples in black-box settings involves proposing function-preserving actions and using a uninterpretable method (*Anderson et al., 2018*) or a random method (*Sharif et al., 2019*) to evade DNNs. Different ways are presented to transform malware without changing its functionality, but only a few articles have explained why their approaches work. Due to the non-linearity of DNNs detectors, they rely on uninterpretable methods or random ways to optimize their transformation. These methods are of little help in designing the malware detector. *Demetrio et al. (2021a)* use the genetic algorithm to generate the adversarial examples. *Sharif et al. (2019)* use the transformation randomly. *Anderson et al. (2018)* use DNNs based reinforcement learning to evade the detector which is also uninterpretable. While DNNs have shown great potential in various domains, the lack of transparency limits their application in security or safety-critical domains. *Arp et al. (2022)* claim that artefacts unrelated to the classified target may create shortcut patterns to separate different classes. Consequently, the DNNs may adapt to the artefacts instead of the original problems. It is important to investigate what these models have learned from malware. An interpretable technique is needed to tell us the most influential features. Most of the existing research on the interpretability of DNNs focuses on image classification and NLP processing (*Ribeiro, Singh & Guestrin, 2016*; *Camburu, 2020*; *Lundberg & Lee, 2017*). To improve the transparency of malware classification, researchers have started to work on the explanation issue of malware classification. To the best of our knowledge, *Coull & Gardner (2019)* are the first to explore this topic. They use various methods such as HDBSCAN, shaply value and byte embeddings to analyze the model. They examine the learned features at multiple levels of resolution, from individual byte embeddings to end-to-end analysis of the model. *Johns (2017)* also examine what DNNs have learned in malware classification by analyzing the activation of the CNN filter. They suggest that a CNN-based malware detector could find meaningful features of malware (*Johns, 2017*). *Demetrio et al.*'s *(2019)* work is the closest one to our research. They use integrated gradients to find the most important input and point out that the MalConv model does not learn the key information in the PE file header according to the interpretability analysis. They devise an effective adversarial scheme based on the explanation. *Rosenberg et al. (2020)* made use of the explainable techniques to train a surrogate neural network to represent the attacked malware classifier. They attack the surrogate model instead. Different from these articles, our function-preserving measures are able to process more types of segments and fewer examples are needed.

## TECHNICAL APPROACH

In this section, we discuss the technical approach behind our framework. First, the general algorithm is described. Then, we introduce the approximating boundary model for fitting the DNNs. Next, the data augmentation module are present. Finally, how we create adversarial examples are explained in detail. Throughout the article, we use the following notations. $m$ refers to the original malware, $f(m)$ refers to the output of the DNNs detector (*e.g.*, class probabilities or logits). We use $g(m) = \vec{m} \circ \vec{w}$ to approximate $f(m)$. $\vec{m}$ is the

interpretable data representation vector. $\vec{w}$ is the weight of the linear equation. $\tilde{m}^i$ refers to the perturbed malware in the *i-th* round of the algorithm and $\tilde{m}^i_j$ is the *j-th* perturbed example in each round. $r_j$ is the perturbation that we make.

## Instance attack

Portions of this text were previously published as part of a preprint (*RuiJin et al., 2022*). In the field of adversarial examples for binary code in a black-box setting, the transformation can only be limited to a few discrete actions as mentioned in Section: Function Invariant Transformation 6 because of the semantic invariance constriction. The transformation is discrete and difficult to optimize with gradient-descent methods. Therefore, interpretable models are being exploited to make adversarial attacks. Adversarial attacks can roughly be divided into three categories: gradient-based, transfer-based, and score-based (or decision-based) attacks (*Brendel, Rauber & Bethge, 2018*). Our method is somewhat similar to the transfer-based method. Traditionally a transfer-based method trains a surrogate neural network model on a training set that is believed to accurately represent the attacked malware classifier such as *Rosenberg et al. (2020)*. To train a surrogate model, traditional methods require a large number of examples. This is often impossible in practice. The framework for training our surrogate model is distinct from other models in that it only requires a single sample to be trained. Our framework fits a specific example, not the detection model. To accomplish this, we train a surrogate model that is designed to precisely represent the specific instance being fitted. A locally interpretable algorithm is used in training the surrogate model. For example, for a DNN detector $f$ and a malware $m_0$, we can train a surrogate model $g(m^1_0) = f(m^1_0), g(m^2_0) = f(m^2_0), \cdots$. But for another malware $n$, $g(n_0) \neq f(n_0)$.

A locally generated linear function $g$ is used to find the important features of the binary, then the function invariant transformation mentioned in Section: Function Invariant Transformation 6 is used to remove the features. Since only one binary is required, we name it Instance Attack.

## General framework

Our model works in a black-box setting. We assume that we have no access to the model parameters and the data set. We don't have any idea about the structure of the classification model and the distribution of the data set. Only one binary instance is given. There is no limit to the number of queries. We can transform the instance arbitrarily. Our target is to generate a new binary to mislead the classifier. After the transformation, we have to guarantee the functionality of the program. The basic intuition of our framework is to approximate the result of the specific example with a linear function and make the perturbation towards the approximate decision boundary iteratively. The whole procedure works in rounds, where each round consists of three steps. In the first step, data augmentation is used to generate a large number of new samples from the original binary. Then a linear model is used to fit these samples. Here, the FastLSM algorithm will be used (described in detail in Section: Interpretable Data Representations and Segmentation Algorithm) to fit the malware detector $f$ with a linear function $g$. The second step is to

approximate the decision boundary by solving the linear equations $g = benign$, and transforming the most important part of the malware accordingly to make $g(\tilde{m}^i) = benign$. For the function invariant restriction of the binary file, $\tilde{m}^i$ is transformed to $\tilde{m}^{i+1} = \tilde{m}^i + r^i$ with the function invariant transformation we propose in Section: Function Invariant Transformation. We finally query the black-box detector to get $f(\tilde{m}^{i+1})$. If the result of $f(\tilde{m}^{i+1})$ is still malware, we move on to the next iteration or we stop the algorithm if the maximum number of iterations is reached. Algorithm 1 shows the pseudocode of the whole procedure.

### *How interpretability is applied*

In traditional detection models, the initial step involves selecting features, followed by designing the detection algorithm. In contrast, DNNs are end-to-end models, they eliminate the need for explicit feature extraction. They can directly input raw data into the model and learn the desired outcomes. DNNs are often referred to as featureless models due to their ability to learn features internally. Similarly, existing attack models that target DNNs do not possess knowledge of the feature weights. These attack models launch direct attacks without this crucial information. While direct attacks may occasionally be successful, they fail to provide insights into understanding deep detection models. In contrast, our approach focuses on acquiring the features employed in deep detection models or identifying the associated feature weights. By modifying features with higher weights, we aim to achieve the attack objective. We utilize the instance attack model to calculate the weights of different parts (superpixels) within the binary. If a superpixel possesses a high weight in this model and is associated with code segments or data segments, we proceed with targeted attacks. These steps are iterated, resulting in the displacement of superpixels. Through this process, we are able to gain insights into the functioning of the DNN model and effectively execute targeted attacks.

## Formalizing the model

Adversarial examples are variants of normal examples by adding some imperceptible perturbations. The adversarial examples cause the detection model to misclassify examples with high confidence. *Carlini & Wagner (2017)* model the adversarial examples as a constrained minimization problem:

$$min \quad D(\tilde{m}^i, \tilde{m}^{i+1}) \tag{1}$$
$$s.t \quad f(\tilde{m}^{i+1}) = benign \tag{2}$$

$\tilde{m}^i$ is fixed and the object is to find the perturbation $\tilde{m}^{i+1}$ that can minimize $D(\tilde{m}^i, \tilde{m}^{i+1})$ and further leads to the evasion. $D(\tilde{m}^i, \tilde{m}^{i+1})$ is a distance function and the perturbation is subject to $f(\tilde{m}^{i+1}) = benign$. Since the identifier is a black-box model, it is difficult to find a solution for the original function $f$. *Carlini & Wagner (2017)* proposed to solve a simple objective function $G$ instead, $G(\tilde{m}^{i+1}) = benign$ if and only if $f(\tilde{m}^{i+1}) = benign$.

---

**Algorithm 1** General algorithm.

**INPUT:** a malware $m$, a classifier $f$, a linear equation $g$, the approximation algorithm FastLSM(), a functional invariant transformation function Tran(), address of most weighted data *[start,end]*

**OUTPUT:** new malware $\tilde{m}^i$

1:  **while** $i < maxiteration$ **do**
2:      $g(\tilde{m}^i) \leftarrow FastLSM(\tilde{m}^i, f())$
3:      $start, end \leftarrow solving\ g(\tilde{m}^i) = benign$
4:      $\tilde{m}^{i+1} \leftarrow Tran(start, end, \tilde{m}^i)$
5:      **if** $f(\tilde{m}^{i+1}) == malware$ **then**
6:          $i \leftarrow i + 1$
7:      **else**
8:          **return**        *success*
9:      **end if**
10: **end while**
11: **return**        *false*

---

$$min \quad D(\tilde{m}^i, \tilde{m}^{i+1}) \tag{3}$$
$$s.t \quad G(\tilde{m}^{i+1}) = benign \tag{4}$$

In malware detection, these formulas are also utilized to find adversarial perturbations $\tilde{m}^{i+1}$ for the original binary $\tilde{m}^i$ that target a class $f_{benign}$. In a black-box environment, finding an object function $G$ such that $G(\tilde{m}^{i+1}) = benign$ if and only if $f(\tilde{m}^{i+1}) = benign$ is an overly strong requirement. Here the local explanation $g$ is used instead. If $f(\tilde{m}^{i+1}) = benign$ is true, then $g(\tilde{m}^{i+1}) = benign$ is established, but the opposite is not necessarily true. When $g(\tilde{m}^{i+1}) = benign$ is true, $f(\tilde{m}^{i+1})$ may not be *benign*. The optimization could be converted to the following problem:

$$min \quad D(\tilde{m}^i, \tilde{m}^{i+1}) \tag{5}$$
$$s.t \quad g(\tilde{m}^{i+1}) = benign \tag{6}$$

Equation (6) can easily be solved to get the minimal perturbation $r$ from $\tilde{m}^i$ to $\tilde{m}^{i+1}$, since $g$ is a linear equation. Because $g(\tilde{m}^i + r)$ is the approximation of $f(\tilde{m}^{i+1})$, we must incorporate the perturbation $\tilde{m}^{i+1}$ back into the original classifier $f$ to get the accurate value. If $f(\tilde{m}^{i+1}) = benign$, then we stop the algorithm or we should recompute the linear approximation of $f(\tilde{m}^{i+1})$. We continue to repeat this process until the evasion is successful or the maximum number of iterations is reached.

## Local linear explanations of malware detection

Here, how we build the linear function $g$ are explained. Our method is inspired by Local Interpretable Model-Agnostic Explanations (LIME) (*Ribeiro, Singh & Guestrin, 2016*) and

Locally Linear Embedding (LLE) (*Roweis & Saul, 2000*). Deep learning algorithms provide highly satisfactory results. However, their decision procedures are non-linear and the important parts of the input data cannot be featured out directly. When one instance are given to bypass the classifier, we try to infer how the detector behaves around a specific instance by querying the detector for the results of different transformed examples. The data augmentation method are used to produce the adversarial perturbations. As claimed in *Roweis & Saul (2000)*, it is assumed that binary files can be represented by points in a high-dimensional vector space. The binary and its transformations lie on or close to the locally linear path of the manifold. In this context, it is assumed that the coherent structure between the binary and its variants leads to strong correlations, which can be characterized by linear coefficients. The use of a linear model is considered simple and interpretable, as it allows for a clear understanding of the relationship between the different features of the binary and its variants.

$$\xi(m) = \text{argmin}_{g \in G} \mathscr{L}(f, g, \Pi_m) + \Omega(g) \tag{7}$$

Equation (7) describes how to solve this problem, where $f$ denotes a DNNs detector, and $g$ is an interpretable model to approximate $f$ without knowing its parameters. In classification, $f(m)$ produces the probability (or a malware indicator) that $m$ belongs to a certain category. If $g$ is a potentially interpretable function, and $\Omega(g)$ measures the complexity of the explanation. $\mathscr{L}(f, g, \Pi_m)$ measures how unfaithful $g$ is in approximating $f$ in the locality defined by $\Pi_m$. To ensure the interpretability and local closeness, $\mathscr{L}(f, g, \Pi_m)$ should be minimized and $\Omega(g)$ should be low enough. The lower the $\Omega(g)$, the easier it is for humans to understand the model.

In this article, $G$ could be the class of linear models, such that $g(m) = \overrightarrow{w_g} \circ \vec{m}$. It is stated that an interpretable representation $\vec{m}$ of $m$ can be obtained directly, the specific rule is described in the following section. it is defined that $\Pi_m(\tilde{m}_i^i) = D(\tilde{m}_i^i, \tilde{m}_0^i)$, where $D$ is some kind of distance function, for example, the *L2*-norm distance. $\tilde{m}_0^i$ is the specific example and $\tilde{m}_j^i$ is the perturbed example in each round. We carefully choose some perturbed examples $\tilde{m}_1^i, \tilde{m}_2^i, \tilde{m}_3^i \cdots, \tilde{m}_j^i$ within each round. For information on the method of perturbation, see Section: Data Augmentation Module 6. $\mathscr{L}$ could be a locally weighted square loss as defined in Eq. (9). In this way, the function has been converted into a linear function fitting problem.

$$g\left(\tilde{m}_j^i\right) = \overrightarrow{w_i} \circ \vec{m}_j^i \tag{8}$$

$$\mathscr{L}(f, g, \Pi_m) = \sum_j \Pi_m(\tilde{m}_i^i)\left(f\left(\tilde{m}_j^i\right) - g\left(\tilde{m}_j^i\right)\right)^2 \tag{9}$$

Given a malware instance $\tilde{m}^i$ and $f$, this problem is a typical ordinary least square problem. It is suggested that examples around $\tilde{m}^i$ can be sampled by drawing non-zero elements uniformly at random to get the value of $\tilde{m}_j^i$. $f(\tilde{m}_j^i)$ could be obtained by querying the classifier. $\vec{m}_j^i$ is the interpretable data representation that can be easily calculated. By

solving these functions, the weight $\vec{w_i}$ can be obtained for sample $i$. $g$ is the local explanation model of $f$. $g$ could also be used as the approximate function of $f$.

## Data augmentation module and optimization algorithm

### Data augmentation module

Ablation analysis is often used in evaluating the DNNs model. It is a technique for evaluating machine learning models by discarding certain features (*everybodywiki, 2020*). We adopt similar ideas by discarding certain features of the instance. As our model is instance-based, we do not quantify the DNNs detector, but the important portion of a specific instance. Augmented examples are created from a given instance by discarding (masking) certain features. Then a linear function $g$ is used to fit the DNNs around these examples, and the weights for each feature are calculated. Computing the weights of all the bits is time-consuming. Given a file of length $LM$, it would take $o(LM^3)$ time to find the most important bit using the Least Square Method. To improve efficiency, two optimization mechanisms are proposed. First, the superpixel are used as the basic unit of interpretable representation. Then, the FastLSM algorithm are introduced to reduce the computational complexity.

### Interpretable data representations and segmentation algorithm

Superpixels were used in image segmentation originally (*Ghosh et al., 2019*). Common image segmentation algorithms include quick shift, felzenszwalb, slic. In this article, superpixels are the results of a binary file over segmentation. We could use the tools to disassemble the binary file, such as IDA, Binary ninja. Superpixels are the basic function blocks returned by disassembly tools. The Capstone disassembler framework are used in our experiment (*Capstone, 2021*). A basic block is a straight-line sequence of codes with only one entry point and only one exit. Or the examples could just be segmented by their offset. For example, a large binary file of size 200 KB could be divided into ten parts, each of which would be 20 KB in size. As shown in Fig. 1, there are three superpixels returned by the disassembly tool. The staed rt offsets of these superpixels are $0x10004675h$, $0x1000469Ah$, and $0x100046A1h$. The lengths of the three superpixels are $0x25h$, $0x7h$, and $0x8h$ respectively. Interpretable data representations $\vec{m}$ and data features $m$ are different. Features $m$ in the range $m\varepsilon R^L$ are the ground truths. An interpretable representation $\vec{m}$ is a binary vector indicating the "presence" (denoted by **1**) or "absence" (denoted by **0**) of a patch of codes and its range is $\vec{m}\varepsilon\{0,1\}^l$. $L$ is the length of the features and $l$ is the length of the interpretable representations. Given a file whose content is "0x1122", "0x11" and "0x22" are the two super pixels and "11" is their interpretable representation. $f("0x1122")$ is equal to $g("0x1122") = w_1 * 1 + w_2 * 1$. Sampling around a specific offset in a binary file, such as "0x1122", can help to generate the corresponding interpretable representation of the data. For example, if "0x1122" is transformed to "0x0022", its interpretable representation is transformed to "01". Table 1 shows the examples in detail.

```
10004675 loc_10004675:                              ; CODE XREF: DllInstall+81
10004675                                             ; DllInstall+1C↑j
10004675                     cmp     [ebp+bInstall], 0
10004679                     mov     ecx, offset dword_10014334
1000467E                     jz      short loc_100046A1
10004680                     push    esi
10004681                     push    1
10004683                     call    sub_10004520
10004688                     mov     esi, eax
1000468A                     test    esi, esi
1000468C                     jns     short loc_1000469A
1000468E                     push    1
10004690                     mov     ecx, offset dword_10014334
10004695                     call    sub_100045B0
1000469A
1000469A loc_1000469A:                               ; CODE XREF: DllInstall+30
1000469A                     mov     eax, esi
1000469C                     pop     esi
1000469D                     pop     ebp
1000469E                     retn    8
100046A1 ; ---------------------------------------------------------------
100046A1
100046A1 loc_100046A1:                               ; CODE XREF: DllInstall+2E
100046A1                     push    1
100046A3                     call    sub_100045B0
100046A8                     pop     ebp
100046A9                     retn    8
```

**Figure 1  An example of superpixels.**

**Table 1  An example of interpretable data representation.**

| Interpretable representation | 00 | 01 | 10 | 11 |
| --- | --- | --- | --- | --- |
| Original feature | 0x0000 | 0x0022 | 0x1100 | 0x1122 |

*Fast least square method*

In Algorithm 1 referred, it is necessary to find the minimum perturbation at each round. Sometimes the weight of the top-$k$ important superpixels is needed instead of each superpixel. *Fleshman et al. (2018)* introduce a segmentation algorithm to find the most important superpixels. We extend it and combine it with our local explanation algorithm. It is named as the Fast least square method (FastLSM), because it can reduce the computational complexity to $o(log(LM))$ to compute the weight of the most important superpixel.

   The steps of FastLSM are as follows. The entire binary are selected as the base superpixel. First, the superpixel are divided into $n$ different superpixels. each super pixel is occluded respectively with zero to generate variants and each variant is analyzed by the DNNs $f$. Second, we choose the variant that results in a larger drop in classification confidence. Third, if the length of the superpixel with a larger drop is smaller than a specific value $\beta$, it is set as the base superpixel and the algorithm starts over from the beginning. There are different ways to occlude the superpixel. This can be done by replacing it with random values, null values, or adversarial values (*Fleshman et al., 2018*). We will discuss how to determine the hyperparameter $\beta$ in the Miscellaneous section. Figure 2 illustrates the overarching process of FastLSM, depicting the general flow and steps involved in the algorithm. The Algorithm 2 provides a comprehensive breakdown of

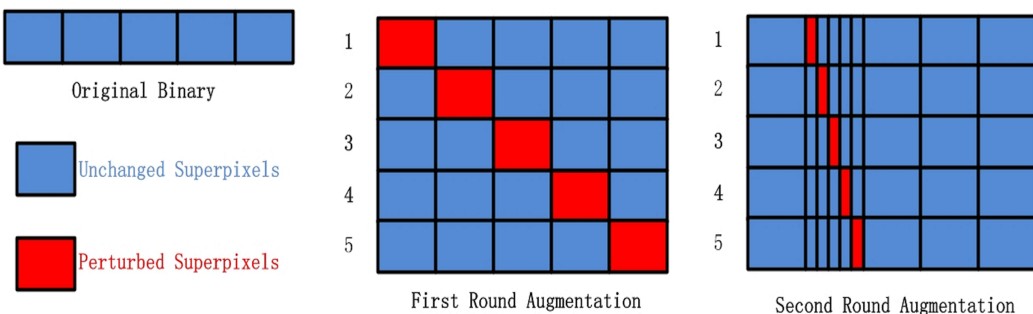

**Figure 2  The data augmentation algorithm used in our experiments.**

the fundamental stages comprising FastLSM, elucidating the specific actions and operations undertaken within the algorithmic framework.

## Function invariant transformation

The semantics of binaries hinder the direct transplantation of existing traditional adversarial learning methods. Even changes as small as one bit can disrupt the original syntax of the binary and may cause it to malfunction or even crash. For example, if the characters are changed from 53 to 52 in the binary file, at the assembly level it means that push ebx is changed to push edx and the function of the generated adversarial example would be invalid. Because of the function preserving constraint, the transformations that we use are limited to those that preserve the function of the binary. In this article, three families of transformations are used. The first transformation is an appending algorithm that applies the evasion by appending adversarial bits at the end of the original files (*Suciu, Coull & Johns, 2019*). These bytes cannot affect the semantic integrity. Appending bytes to inaccessible regions of the binary may be easy to detect and could be sanitized statically (*Sharif et al., 2019*). The second transformation that we use is named code displacement (Disp), which was proposed by *Koo & Polychronakis (2016)* to break the gadgets originally. *Sharif et al. (2019)* adopt it to mislead DNNs-based classifiers. The general idea of Disp is to move some codes to the end of the binary. The original codes are replaced by a *jmp* instruction at the beginning of the code segment. After filling the newly allocated code segment with the adversarial codes, another *jmp* instruction is immediately appended. The third transformation is an original one named data displacement (DataDisp). Disp can only transform code segments rather than data segments. However, we find that the DNNs sometimes attach importance to data segments as shown in the experiment (Section: Distribution 6). Therefore, we propose DataDisp, an algorithm that could be used to transform data segments (this transformation can also be applied to code segments with some modifications). DataDisp is a practical code diversification technique for stripped binary executables. DataDisp transfers data to the end of the file and uses the *mov* instruction to move the data back before the binary is executed. It starts by adding a new section at the end of the PE file. The original codes to be moved are replaced with adversarial data (*Fleshman et al., 2018*), random data, or null data. After filling the newly allocated segment with the *mov* code, a *jmp* instruction is immediately appended to give

**Algorithm 2 FastLSM algorithm.**

**INPUT:** a malware $m$, a classifier $f$, a target occlusion size $\beta$, $n$ is the number of segments within each iteration

**OUTPUT:** a new malware $\tilde{m}$

1: Split file $m$ into $n$ sections, $splitsize \leftarrow |m \div n|$, size of $ith$ section is $|m(i) = splitsize|$

2: Use LSM to get the weight for each section

3: Find the $jth$ section with the maximum weight

4: $start \leftarrow$ start address of $jth$ section

5: $end \leftarrow$ end address of $jth$ section

6: **while** $splitsize > \beta$ **do**

7:     $max \leftarrow 1, i \leftarrow 1$

8:    **while** $i < n$ **do**

9:       $starttemp \leftarrow (start + splitsize * (i - 1))$

10:       $endtemp \leftarrow (start + splitsize * i)$

11:       $m_i[starttemp:endtemp] \leftarrow$ 0x00, occlusion the ith segment with 0x00

12:       $starttemp \leftarrow (start + splitsize * (max - 1))$

13:       $endtemp \leftarrow (start + splitsize * max)$

14:       $m_{max}[starttemp:endtemp] \leftarrow$ 0x00, occlusion the most weight segment with 0x00

15:       **if** $f(m_i) < f(m_{max})$ **then**

16:         $max \leftarrow i$

17:         $i \leftarrow i + 1$

18:       **else**

19:         $i \leftarrow i + 1$

20:       **end if**

21:    **end while**

22:    $start \leftarrow start + splitsize * (max - 1)$.

23:    $end \leftarrow start + splitsize * (max)$.

24:    $splitsize \leftarrow splitsize \div n$

25: **end while**

26: **return** $\tilde{m}[start : end]$

control back to the original binary. At last, the OEP are changed to the beginning of the newly allocated section. There are a few tips to be noted. If the displaced codes contain important structural information (*e.g.*, edata section) we leave them alone.

As shown in Fig. 3A, the original data at $0x402A4E$ is the string *fail*, and is replaced by $0x00$. The Original Entry Point (OEP) address is displaced to $0x47E14B$ in Fig. 3B. Then the address $0x402A4E$ is stored in *EAX*. The new code will reconstruct the original file using *mov*. After the reconstruction, it will *jmp* back to the original OEP.

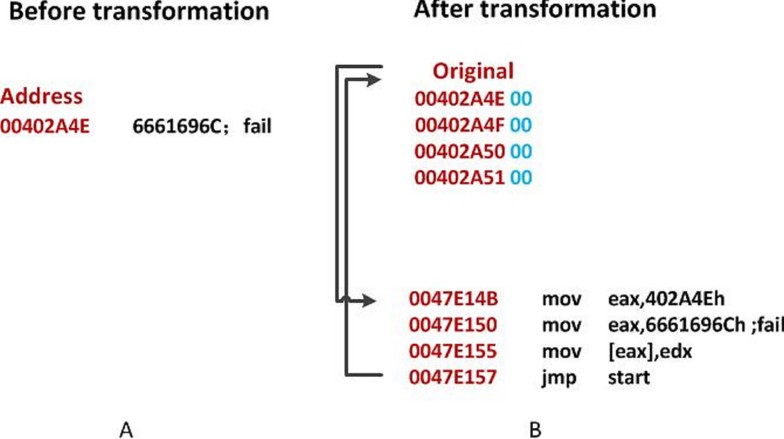

**Figure 3 An illustration of DataDisp transformation.** (A) Before transformation. (B) After transformation.

## EVALUATION

This section presents comprehensive empirical evidence for the adversarial theme. First, we describe the datasets and the DNN detectors in detail. We also discuss the interpretational analysis on the experimental data. Then, different methods of transformations are presented. Finally, our model is compared with three other methods. The results demonstrate that our adversarial example generation model is trustworthy.

### Datasets and malware detector

All experiments were conducted on an Ubuntu 18.04 server with an Intel Xeon CPU and 64 GB of RAM. The computer was equipped with Python 3.7, PyTorch, and an NVIDIA GTX3090 Graphics processing unit. A mixed data set is used. We resorted to a publicly available dataset to collect malware. And benign binaries are collected by ourselves. The malware binaries were adopted from the Kaggle Microsoft Malware Classification Challenge (*Ronen et al., 2018*). This dataset contains nine different families of malware. Even though there are no benign ones, we still used the dataset to train our model. It is based on the following considerations. First, most previous works (*Anderson & Roth, 2018*; *Krčál et al., 2018*; *Raff et al., 2018*) used proprietary datasets and some other public datasets contain only packed data (*Vigna & Balzarotti, 2018*). The dataset of *Noever & Noever (2021)* contains only processed data. Although *Yang et al. (2021)* also offer sufficient raw files, they do not provide benign files either, for copyright reasons. Second, over fifty articles had cited this dataset (*Ronen et al., 2018*), which was the *de facto* standard for malware classification. Our dataset of benign binaries was collected from a newly created Windows 7 machine. We used two specific tools, Portable apps and 360 package manager, to install a total of 180 different packages. To ensure that our dataset was representative, we also included popular files such as Chrome, Firefox, and notepad++. In addition, we collected packages that were likely to be used by academics (such as MiKTeX and

**Table 2 The number of binaries for training, validating and testing the DNNs.**

| Class | All | Train | Validation | Test |
|---|---|---|---|---|
| Malware | 10,868 | 8,708 | 1,080 | 1,080 |
| Benign | 9,814 | 7,814 | 1,000 | 1,000 |

MATLAB), developers (such as VSCode and PyCharm), and document workers (such as WPS and Adobe Reader). By including a diverse range of packages, our dataset provides a comprehensive representation of commonly used software across various domains. We selected 9,814 benign binaries, most of them were smaller than 2 MB. The files are divided into the train, validation, and test sets as shown in Table 2.

Two DNN classifiers were chosen from those mentioned in Section: Background and Related Work. Both classifiers were given raw byte binaries. The first classifier, named AvastNet, is a four-layer convolutional neural network with four fully connected layers. It receives inputs of up to 512 KB and was proposed by *Krčál et al. (2018)*. The second DNN model is called MalConv and was proposed by *Raff et al. (2018)*. Its network structure consists of two 1-D gated convolutional windows with 500 strides, and it receives inputs of up to 2 MB. To evaluate the performance of these models, we split all the files into three sets: training, test, and validation. Both of these DNNs achieved accuracies above 95% on our datasets. The classification results of these two DNNs are shown in Table 3.

Because there were no PE headers in the Microsoft dataset, disassembling the binaries to validate our algorithm was not possible. Consequently, we turned to VirusShare and downloaded malware samples from the nine malware families that were not present in the Microsoft dataset. VirusShare is an open repository of malware samples with labels. We obtained 88 binaries that belonged to the same nine families but did not appear in the training set. These samples are then evaluated against MalConv and AvastNet separately. The results are shown in Table 4, where 68 of them were identified as malware by MalConv. We also sampled 88 benign binaries for the test, 10 of which were misclassified by MalConv. AvastNet marked 72 of them as malware, as shown in Table 4. We also sampled 88 benign binaries for testing, eight of which were misclassified by AvastNet.

In addition to the two DNNs that we trained, we evaluated our attacks using a publicly avaible model Endgame (*Anderson & Roth, 2018*). It is a gradient boosed decision tree (GBDT) model using LightGBM with default parameters (100 trees, 31 leaves per tree), resulting in fewer than 10K tunable parameters. The dataset of Endgame includes features extracted from 1.1M binary files: 900K training samples (300K malicious, 300K benign, 300K unlabeled). As there is no raw file available in Endgame dataset, we turned to VirusShare. For our evaluation, we utilized the identical set of 176 binaries that were previously tested in MalConv and Avast to assess the performance of the Endgame model. Out of these 88 malware binaries, Endgame successfully identified 77 of them as malware. Additionally, we included 88 benign binaries in our test set, out of which Endgame erroneously misclassified five as malware.

**Table 3 The DNN's performance.**

| Classifier | Accuracy | | |
|---|---|---|---|
| | Train | Validation | Test |
| MalConv | 98.8% | 97.8% | 96.1% |
| AvastNet | 99.8% | 97.2% | 97.5% |

**Table 4 Binaries used to test our model.**

| Class | Model | All | As malicious | As benign |
|---|---|---|---|---|
| Malware | MalConv | 88 | 77.3% | 22.7% |
| Benign | MalConv | 88 | 11.4% | 88.6% |
| Malware | AvastNet | 88 | 81.2% | 18.8% |
| Benign | AvastNet | 88 | 9.1% | 90.9% |
| Malware | Endgame | 88 | 87.5% | 12.5% |
| Benign | Endgame | 88 | 5.7% | 94.3% |

## Weight analysis on superpixels

### Distribution

We used the least square algorithm to obtain the weights of different parts of the binaries and conducted a statistical analysis to understand the black-box classifier. The sampled files were divided into superpixels that were smaller than 8 KB and larger than 4 KB, and we summarized the corresponding weights of these superpixels for the MalConv and AvastNet classifiers. As shown in Table 5, 90% of the superpixels had a weight of less than 0.1, and only about 5% of the superpixels were between 0.1 and 0.2. A total of 0.5% of the superpixels had a weight greater than 0.2. We can even estimate their distribution. We plot the weight of different superpixels of the malware on MalConv, as shown in Fig. 4. In general, although the sum of the weights of all the superpixels must be greater than 0, the weight distribution conforms to the normal distribution, and the mean is approximately zero. Figure 4 shows the result of the weight analysis on MalConv, and the analysis on AvastNet looks similar.

According to the weight distribution diagram, we can conclude that most of the contents have little influence on the result. There are only less than one percent of the superpixels that have a significant impact on the classifier, whose weight is greater than 0.2. We could use the data as adversarial data in the following experiments. Some superpixels with higher absolute weight are listed below. The data that is shown in Fig. 5B has a negative impact on the MalConv classifier. It is a URL for digicert (digicert.com). This website is obviously not malicious. The presence of these codes in binaries can increase the probability of being classified as benign. The data in Fig. 5A plays positively to AvastNet, this figure is the disassembly result of IDA Pro. We can see that the code is the import table of a PE file. Many of the functions in the table are related to malicious behavior with high
**Table 5 Percentage of superpixels with different weights.**

| Weight | [0, 0.01] | [0.01, 0.1] | [0.1, 0.2] | [0.2, 1] |
|---|---|---|---|---|
| MalConv | 43.2% | 50.8% | 5.5% | 0.5% |
| AvastNet | 44.2% | 50.2% | 5.2% | 0.4% |

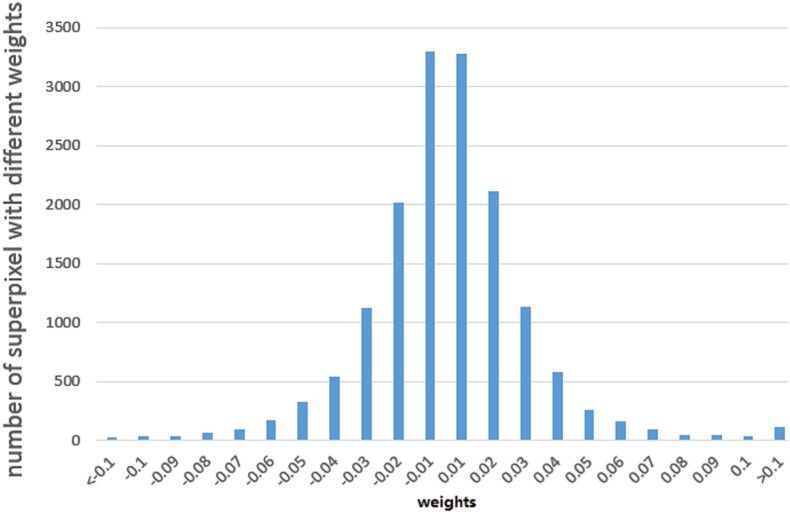

**Figure 4 Weight distribution diagram under the MalConv classifier.** It shows that the weight distribution curve of the malware follows the normal distribution.

probabilities, such as the isdebuggerpresent which is the API that is often used by malware to resist reverse analysis.

We also analyzed the malware containing malicious APIs with the lightGBM model Endgame which is introduced by *Anderson & Roth (2018)*. The lightGBM model was not trained on the same dataset as our model. Although the output of lightGBM was 0.8388964 which implied that there was a high probability that the file was malware. But by analyzing the file with our interpretable model, we can see that the model gave most weight to the file's PE header. We show the weight and offset of the three most weighted superpixels in Table 6. We could conclude that the lightGBM model makes decisions according to the header features.

### Proportion of code segment weight

We also examined the proportion of code segments in the total score generated by the classifiers. To analyze the results, we employed an explanation-based model. The weight of the code segments are calculated by adding the weight of all the superpixels that belonged to the code segments. Although it was not strictly defined, it corresponded to the code/text section of the binaries (*Microsoft, 2021*). However, the author of the malware could change the name of the code segment at will. By using the explanation model, we could get the weight of all the sections (bss, edata, idata, idlsym, pdata, rdata, reloc, rsrc, sbss, sdata, srdata, code/text). We computed the weight of code/text for all binaries and presented the

**Figure 5 The data with varying weights exhibit contrasting effects on the results.** On the right (B), the data includes a benign URL, digicert.com that can raise the probability of being classified as benign. Conversely, the data on the left (A) demonstrates a positive effect. Numerous functions listed in the table exhibit high probabilities of being associated with malicious behavior, such as the isdebuggerpresent API, frequently exploited by malware to evade reverse analysis. This increases the likelihood of being categorized as malware.

**Table 6 The weight and offset of the three most weighted superpixels of the malware under the lightGBM.**

| Offset | 0x0000–0x1000 | 0xe000–0xf000 | 0xc000–0xd000 |
|---|---|---|---|
| Weight | 0.93 | −0.0277 | −0.0272 |

**Note:**
0x0000–0x1000 is the address of the PE header. The model gave too much weight to the file's PE header, we concluded that the lightGBM model makes a decision according to false causalities.

CDF in Fig. 6. The CDF reveals that the weight of code sections amounts to roughly 50% in half of all the binaries. Although this was only an estimation, the weight of the code segments must be limited. We concluded that code sections only account for part of the weight. Due to the fact that not all data segments can be transformed by Disp, it is highly probable that the success rate will be low if the Disp algorithm is used alone.

## Randomly applied transformations

In order to study the influence of the location of the transformed content and the type of the transformed content on the success rate of adversarial examples, we evaluated whether the randomly applied transformation would lead to evasion of the DNNs. To evaluate the transformations, we created up to 200 variants for each binary. If the detection results of more than one variant changed, the transformation would be considered successful. The binary are divided into superpixels. For code sections, the superpixel was the basic

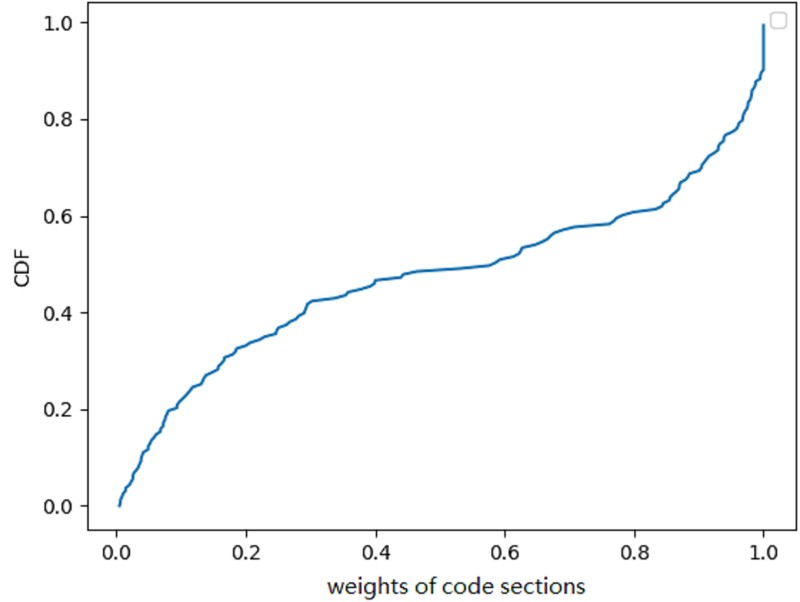

**Figure 6 A cumulative distribution function (CDF) graph depicting the weight of code sections contributing to the decision provides valuable insights into the behavior of a malware detection model.** Approximately 50% of the binary's code sections have little contribution to the final result.

functional block returned by the disassembly tools. For data sections, we divided binaries into 1 KB length superpixels by offset. The concept of "randomly applied transformations" encompassed two aspects. Firstly, whether a particular superpixel in the binary underwent transformation was determined randomly. Secondly, the gap space following the transformation was filled either with adversarial data or random data. Specifically, we designed two experiments. We conducted two distinct experiments to test our approach. In the first experiment, we filled the gap spaces with random data, while in the second experiment, we used adversarial data to fill the gaps. The adversarial data was the data we found in the previous section with a high absolute weight.

We conducted both experiments with the constraint that the size of each binary would not increase by more than 5%, and limited the number of iterations to 200 for both Disp and DataDisp. When Disp and DataDisp were both used randomly with random data, the results showed that three malware binaries were misclassified, and five benign binaries were incorrectly classified as malware when using MalConv. Four malware and six benign binaries were misclassified for AvastNet. The results are easy to explain under our framework, because the weight is under a normal distribution with a mean value of 0 as shown in Fig. 4. If the Disp & DataDisp algorithms are applied randomly, the weight of the transformed binaries is also under a normal distribution and the sum of the weights has a high probability with a mean value of 0. There is a high probability that the adversarial examples will not evade the detector. So we could conclude that it's not that the DNNs are robust against naive Disp transformations as claimed in *Sharif et al. (2019)* but it's just a matter of probability. However, when we filled the gap spaces with adversarial data, the results improved significantly. The adversarial data was selected from the higher-weighted

data we identified in the previous section. Using this approach, we achieved a success rate of 24% for MalConv and 39% for AvastNet, which represented the highest success rates obtained in our experiments.

## Evaluation of the transformations: Disp and DataDisp

In this section, we evaluated the Disp and DataDisp transformations individually using an interpretable model to optimize the procedure. With regards to Disp, we set the maximum displacement budget to 5% and the maximum number of iterations to 200. We filled the gap left by the transformation with adversarial data. The Disp algorithm could achieve a maximum success rate of 59% for MalConv and 45% for AvastNet. With regards to DataDisp, we used a maximum displacement budget of 5% and a maximum number of iterations of 200. Once again, we filled the gap spaces left by the transformation with adversarial data. The DataDisp algorithm yielded a maximum success rate of 53% for MalConv and 35% for AvastNet.

*Sharif et al. (2019)* also tested Disp with a hill-climbing approach. They only moved subsections that had a positive impact on the results. They got a maximum success rate of 24%. Our hypothesis was that this result was due to the fact that Disp was only capable of transforming the code section of a binary file.

## Evaluation on explanation-based adversarial algorithm

In this subsection, we evaluated our explanation-based model by comparing it to other algorithms. We set the maximum displacement budget to 5% and limited the number of rounds to 200. To improve the performance, we used a combination of Disp and DataDisp transformations. We compared our model with three different models, they were Disp with a hill-climbing approach (*Sharif et al., 2019*), genetic padding (*Demetrio et al., 2021a*) and gradient-based attack (*Kreuk et al., 2018*). All of these algorithms, including the explanation-based model, increased the length of the binary by padding different contents at the end of the file. The gradient-based algorithm operated in a white-box setting, using the parameters of the DNNs to calculate the gradient (*Suciu, Coull & Johns, 2019*; *Kreuk et al., 2018*). The gradient-based padding we used was adapted from *Kreuk et al. (2018)* with epsilon 0.5 and iteration 2. The genetic padding was a black-box approach that we adapted from *Demetrio et al. (2021a)* with iteration 10 and population 50. The genetic padding required data randomly sampled from different files. Similar to the genetic padding algorithm, our approach also worked in a black-box setting. However, we improved upon this method by adding binary files with the most weighted data identified in the previous section. The impact of this modification can be seen in the results displayed in Fig. 7.

Although our algorithm was not as effective as the gradient-based model, we observed that it outperformed genetic padding and Disp with the hill-climbing approach. Notably, the gradient-based attack requires model information that may be unavailable in practice. Among all black-box models, our attack model yielded the best performance. However, without budget constraints, our explanation-based model could resulted in a much higher number of misclassified binaries. To comprehensively evaluate our attack model, we also

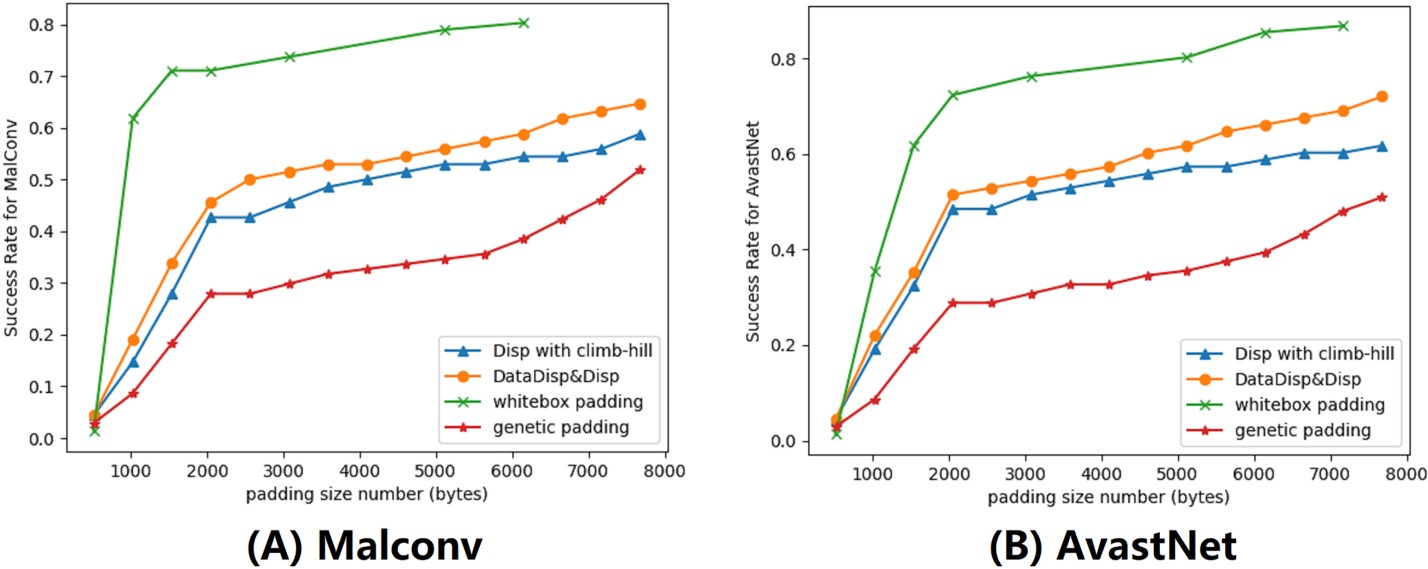

**Figure 7** **We provided a demonstration of various attacking algorithms, where the orange line represents the misclassification rate of our black-box algorithm.** Our approach proved to be less effective than the gradient-based algorithm, but outperformed both genetic padding and Disp in combination with the climb-hill algorithm within a certain range. (A) Malconv. (B) AvastNet.

conducted attacks on the Endgame model. The results demonstrate a maximum success rate of 52%. The Endgame model, which benefits from a larger training dataset, incorporates substantial structural information in its features. These features, unfortunately, are immutable in our attack model, consequently resulting in the relatively poorer performance of our attacks.

## Computational analysis

In our article, we evaluated the performance of our attack on three different models: MalConv and Avast, which are DNN-based models, and the Endgame model, which is a gradient boosted decision tree model. The DNN models have a linear complexity when it comes to training, as it depends on the length of the binary input. On the other hand, the complexity of the Endgame model is determined by the number of trees and the number of leaves per tree. It is important to note that none of these models were developed by us, so we focused on analyzing the computation requirements for generating adversarial examples. The complexity of the attack model consists of two parts: the time required to displace instructions and the time required to query the detector. Displacing instructions randomly within a binary function with $k$ instructions has a time complexity of $o(k)$. If the length of the binary function is $n$, the query time is has $o(log(n))$ time complexity. For the instance attack model, the overall running time of the model is equal to the sum of query time and the time to displacing the instructions, which is $o(log(n) + k)$. Regarding the collected data on actual running conditions, there were 1,805, 1,921 query for attacking MalConv and Avast on average. The attacks took 630, 730 seconds on average respectively

for Malconv and Avast model. The average number of queries was 6,210 with an average time of 1,545 seconds for Endgame.

## Miscellaneous

### Hyperparameters

Throughout this article, we chose 200 as the maximum number of iterations and 5% as the maximum displacement budget, we also used 1 KB as the size of the superpixel of DataDisp in Section: Randomly Applied Transformations. We used these hyperparameters because our method could achieve almost perfect success under this configuration. In *Sharif et al. (2019)* and *Lucas et al. (2021)*, they tested Disp with a hill-climbing approach with similar hyperparameters.

### Integrity of binary

To ensure that the functionality of the binaries was intact after the transformation. Firstly, we selected six different binaries and manually checked their instructions with OllyDbg (*Yuschuk, 2014*). Secondly, we selected 10 different benign binaries and manually checked their functionality by running them on Windows. All the files worked fine. Thirdly, we also used the Cuckoo Sandbox (*Guarnieri et al., 2019*) to test 10 malware programs. One of them collapsed after transformation, and the rest ones functioned normally. We checked the file manually. We found that the binary does not strictly follow the PE format specification. The length of the data segment shown in the file header does not match the actual length.

# DISCUSSION AND FUTURE WORK

In this section, we presented the results of our experiments and highlighted areas for future improvement. We also briefly covered the limitations and basic assumptions of our model.

## Discussion on experiments

We carefully designed multiple sets of experiments to evaluate the effectiveness of our approach. First, we analyzed binary files to identify which content the black-box detector valued and used this information for targeted attacks. Second, we randomly applied transformations in Section: Randomly Applied Transformations but found that these techniques alone were not sufficient to achieve high adversarial results. This demonstrated the importance of both optimization and adoption of adversarial data. Third, we compared our model with others and observed that it did not perform as well as the gradient-based attack model. However, it is important to note that the gradient-based model works in white-box settings and requires model information that may not be available in practice. Our adversarial model utilized Disp & DataDisp transformation methods to transform both data and code segments, resulting in the best performance under black-box conditions.

## Discussion on the model

In black-box attacks, adversaries lack knowledge of the internal workings of the model. To overcome this, adversaries may leverage adversarial classifier reverse engineering (ACRE)

to learn sufficient information for recovering the classifier (*Lowd & Meek, 2005*). Another approach for attacking black-box systems is to train a substitute model using synthetic inputs generated by adversaries (*Szegedy et al., 2014*). This method is based on the assumption that two models with comparable performance solving the same ML task are likely to have similar decision boundaries (*Papernot et al., 2018*). We make a similar assumption that two models with comparable performance around one instance are likely to have similar decision boundaries. So we train a surrogate model around a specific example. Compared with training a surrogate model on the entire data set, the instance-based approach greatly reduces the computational complexity of training and reduces the amount of data required.

Our model requires only a single training instance rather than many. In practice, we often have access to only one or a few examples and need to generate enough targeted adversarial samples. The inputs used to train the surrogate model are transformed solely from the example itself because a coherent structure between the binary and its variants leads to strong correlations. We believe that the information learned from these perturbed instances is specific and targeted. By training a simpler substitute model, we can then use this model as our target for attack. This approach is referred as the instance-based attack. In Locally Linear Embedding(LLE) (*Roweis & Saul, 2000*), each data point is a linear combination of its neighbours. As claimed in LLE (*Roweis & Saul, 2000*), we assume that the binary and its perturbed instances lie on or close to a locally linear path of the manifold. So we can characterize each binary from its neighbours by linear coefficients. This is the key assumption of our article.

After discussing the mathematical basis of the instance attack model, we concluded that our fitting function is necessary but not sufficient. Thus, we must iterate the process many times. Unlike in NLP and image classification, the goal is not to minimize perturbation but rather to maintain the function's integrity. For two semantically identical binary files, their characters may have no resemblance to each other. This is why we need to make many transformations.

## Limitations and future works

Limited by our linear fitting model, our interpretable model is not suitable for some structure-based adversarial transformations such as content shifting (*Demetrio et al., 2021b*; *Anderson et al., 2018*). Our algorithm is instance-based, which means that it needs a lot of queries and calculations to do an adjustment for each example. However, the convergence of the algorithm has not been proven and we have to iterate many times. We use a linear function to fit the classifier. We believe that we could introduce some more complex models such as the local non-linear interpretable model (*Guo et al., 2018*), and the accuracy can be furtherly improved. The combination of global fitting and local fitting frameworks is also worth exploring and the intrinsic dimension of our model could also be discussed (*Pope et al., 2021*). In the field of malicious code detection, conventional models do not typically impose limitations on the number of queries, as scanning a single personal computer often necessitates querying millions of files. However, our model does not possess a distinct advantage in scenarios where query efficiency is paramount. Thus far,

there has been no evident demand for such capabilities in the domain of code analysis. Nevertheless, exploring avenues to significantly reduce the number of queries on the base of *Guo et al. (2019)* remains a promising research direction.

### Potential mitigations

While our model has achieved a commendable success rate, it is crucial to develop mitigation measures that bolster the resilience of malware detection against potential evasion efforts stemming from our attack strategies. Although our model necessitates a substantial number of queries for effective implementation, we do not regard this as an inherently efficacious mitigation approach. Static detection is widely acknowledged as being generally undecidable. However, we posit that the following two techniques can impart a degree of mitigation against our attack model. Firstly, our model primarily focuses on the perturbation of code and data segments. By ascribing higher weights to structural elements such as file headers or structure data entities like input/output functions name, we can augment the accuracy of detection. Secondly, Our model does not affect the dynamic execution of the code. Through the integration of static and dynamic detection methodologies, such attack methods can be proficiently circumvented.

## CONCLUSIONS

Our article introduces a new concept, known as the "instance-based attack," through which we analyzed two DNN-based malware classifiers using an interpretable model. Our analysis revealed key characteristics of these models under black-box conditions, highlighting the critical role played by data segments in determining results. This importance of data segments had not been discussed in related articles before. Additionally, we introduced a novel method to generate adversarial examples, which we call the instance attack. Unlike other methods that insert code in invalid places or transform only code segments, our adversarial model can transform both data and code segments using Disp and DataDisp. Our model achieves state-of-the-art results under black-box conditions, and the results of the instance attack can be verified using domain knowledge. We hope that our work will inspire future research efforts in this area.

## APPENDIX

### Windows portable executable file format

The data we use in this article are all Windows PE files and we take advantage of the format characteristics of the PE files to create adversarial examples. The PE files are derived from the Common Object File Format (COFF), which specifies how Windows executables are stored on the disk. The main file that specifies the PE files is winnt.h, related documents can also be found in *Microsoft (2021)*. There are two types of PE files, one is executable (EXE) file and the other is dynamic link library (DLL) file. They are almost the same in terms of file format, the only difference is that a field is used to identify whether the file is an EXE or DLL. Generally, PE files can be roughly divided into different components. They begin with a MS-DOS header, a Stub and a PE file signature. Immediately following is the PE file header and optional header. Beyond that, section headers and section bodies follow.

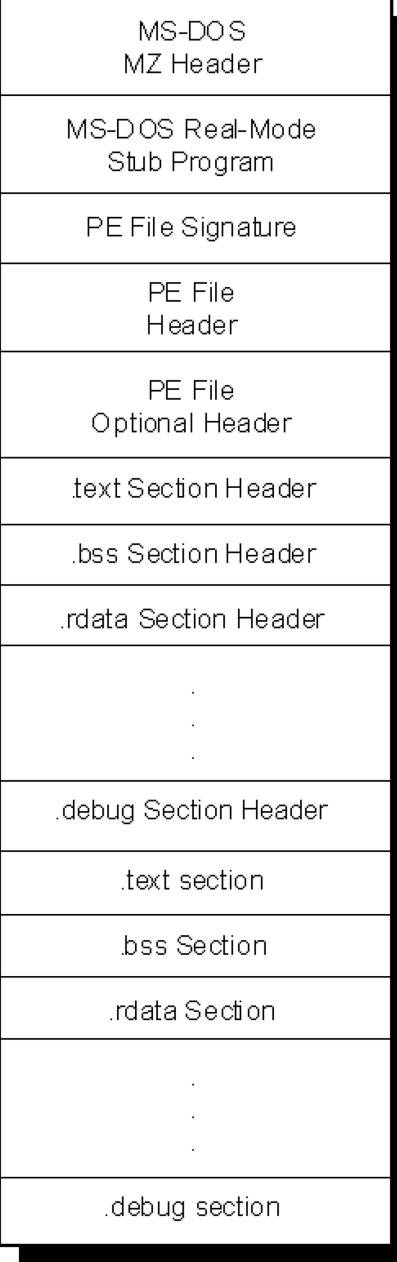

**Figure 8 Structure of a typical PE file image.**

A PE file typically has nine predefined sections named .text, .bss, .rdata, .data, .rsrc, .edata, .idata, .pdata and .debug (*Plachy, 2018*). Figure 8 depicts a typical exemplification of the structure of a PE file. Some binaries do not need all of these sections while others may rename or define the section names according to their own needs. For the alignment reason, the start address of the segment part of PE is often 0x100.

### Funding
This work was supported by the National Key R&D Plan: 2018YFB0805000. The funders had no role in study design, data collection and analysis, decision to publish, or preparation of the manuscript.

### Grant Disclosures
The following grant information was disclosed by the authors:
National Key R&D Plan: 2018YFB0805000.

### Competing Interests
The authors declare that they have no competing interests.

### Author Contributions
- Ruijin Sun conceived and designed the experiments, performed the experiments, analyzed the data, performed the computation work, prepared figures and/or tables, authored or reviewed drafts of the article, and approved the final draft.
- Shize Guo conceived and designed the experiments, prepared figures and/or tables, and approved the final draft.
- Changyou Xing performed the experiments, authored or reviewed drafts of the article, and approved the final draft.
- Yexin Duan analyzed the data, authored or reviewed drafts of the article, and approved the final draft.
- Luming Yang analyzed the data, performed the computation work, prepared figures and/or tables, and approved the final draft.
- Xi Guo performed the computation work, prepared figures and/or tables, authored or reviewed drafts of the article, and approved the final draft.
- Zhisong Pan performed the computation work, authored or reviewed drafts of the article, and approved the final draft.

### Data Availability
The code is available at GitHub and Zenodo:
- https://github.com/iamawhalez/instanceattack.
- iamawhalez. (2023). iamawhalez/instanceattack: intanceattack (intanceattack). Zenodo. https://doi.org/10.5281/zenodo.8242650.
The data is available at Kaggle https://www.kaggle.com/c/malware-classification/data, and detail are described at https://arxiv.org/pdf/1802.10135.pdf.

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
