# Peer review of "Instance attack: an explanation-based vulnerability analysis framework against DNNs for malware detection"

_PeerJ Computer Science, doi:10.7717/peerj-cs.1591_

## Round 0.1 · original submission · Major Revisions

Dear authors,

Thank you for your submission. Your article has not been recommended for publication in its current form. However, we do encourage you to address the concerns and criticisms of the reviewers and resubmit your article once you have updated it accordingly.

Best wishes,

Reviewer 1 ·

Basic reporting

This paper presents a novel attack method, known as the instance-based attack, designed to target Deep Neural Networks (DNNs) used for malware detection. The authors conduct an analysis of DNN-based malware classifiers using this attack method, uncovering the significant role that data segments play in the results. In addition, they introduce a new method for generating adversarial examples, achieving state-of-the-art results under black-box conditions.

The paper provides a fresh perspective, highlighting the crucial role of data segments in malware classification, a viewpoint that has not been given much attention in previous research. The introduction of the instance-based attack as a new method and the demonstration of its effectiveness are noteworthy. The paper's objective is clear, and its structure is well-organized. The reviewer believes that the paper is worthy of publication. However, the authors could enhance the paper by considering the following comments.

The authors acknowledge a significant limitation of the proposed algorithm: it is instance-based, necessitating numerous queries and computations for fitting each example. Given that black-box attacks require query efficiency (as per arXiv:1905.07121) due to their susceptibility to query limiting, this is a critical limitation. However, the paper does not sufficiently discuss strategies to mitigate this limitation, despite the careful consideration given to avoiding the constraints imposed by the use of a linear function. The authors should address this point more thoroughly.

Experimental design

see 1

Validity of the findings

see 1

Reviewer 2 ·

Basic reporting

This paper introduces a new concept called the "instance-based attack" and uses it to analyze two deep neural network (DNN)-based malware classifiers. The analysis reveals key characteristics of these models under black-box conditions, highlighting the critical role played by data segments in determining results. Additionally, the paper introduces a novel method to generate adversarial examples, which they call the instance attack. The authors argue that thier approach can be used to generate sufficient data for training a simple and interpretable model for malware detection, and that providing explanations for the detection model can improve its performance. Overall, the paper provides a framework for explanation-based vulnerability analysis against DNNs for malware detection.

General Comments
The paper provides a clear and concise introduction to the concept of instance-based attack and its potential applications in malware detection.
The authors use a combination of data augmentation strategies and a simple interpretable model to generate sufficient data for training a malware detection model. This approach is shown to be effective in generating adversarial examples that can mislead DNN-based detection systems.
The paper highlights the importance of data segments in determining the performance of DNN-based malware classifiers. This is an important insight that can help improve the robustness of these systems.
The authors provide a detailed analysis of the performance of two DNN-based malware classifiers under black-box conditions. This analysis reveals key characteristics of these models and highlights the importance of providing explanations for the detection model.

Experimental design

The authors only evaluated their approach on two DNN-based malware classifiers. While these are popular models in the field, it would be interesting to see how the proposed approach performs on other types of classifiers or on more complex datasets.
The authors do not provide a detailed analysis of the interpretability of their proposed model. While they argue that their model is simple and interpretable, it would be useful to see a more detailed analysis of how the model makes its decisions and how these decisions can be explained to end-users.
The authors do not provide a detailed analysis of the computational requirements of their approach. While they mention the hardware and software used in their experiments, it would be useful to see a more detailed analysis of the computational resources required to generate adversarial examples and train the proposed model.
The authors do not provide a detailed analysis of the limitations of their approach. While they show that their approach is effective in generating adversarial examples, it would be useful to see a more detailed analysis of the types of attacks that their approach is vulnerable to and how these attacks can be mitigated.

Validity of the findings

* The authors only evaluate their approach on a limited set of malware and benign binaries. While they argue that their dataset is diverse, it would be useful to see how the proposed approach performs on larger and more diverse datasets.

Additional comments

* The paper focuses primarily on malware detection, which may not be sufficient for detecting more complex and dynamic malware. It would be interesting to see how the proposed approach performs in combination with other techniques, such as dynamic analysis or behavior-based detection.

---

## Round 0.2 · Minor Revisions

Dear authors,

Your article has a few remaining issues especially on Figure 5. We encourage you to address the concerns and criticisms of the reviewer and resubmit your article once you have updated it accordingly.

Best wishes,

Reviewer 1 ·

Basic reporting

The authors carefully revised the manuscript according to the reviewers' comments and suggestions.

Experimental design

see 1

Validity of the findings

see 1

Reviewer 2 ·

Basic reporting

I suggest authors to change/ remove Figure 5. 1) It is not clear 2) It doesn't add any value.

Experimental design

NIL

Validity of the findings

NIL

Additional comments

NIL

---

## Round 0.3 · accepted · Accept

Dear authors,

Thank you for the revision. The paper seems to be improved in the opinion of the reviewers. The paper is now ready to be published.

Best wishes,